# The Importance of In Vivo Reflectance Confocal Microscopy in a Case of Desmoplastic Melanoma

**DOI:** 10.3390/life14050574

**Published:** 2024-04-30

**Authors:** Miruna Ioana Cristescu, Liliana Gabriela Popa, Elena Codruta Cozma, Ana Maria Celarel, Valentin Tudor Popa, Mihai Lupu, Vlad Mihai Voiculescu

**Affiliations:** 1Elias University Emergency Hospital, 011461 Bucharest, Romaniaana-maria-stefania.malciu@rez.umfcd.com (A.M.C.); vlad.voiculescu@umfcd.ro (V.M.V.); 2Department of Dermatology, Carol Davila University of Medicine and Pharmacy, 050474 Bucharest, Romania; 3Department of Pathophysiology, University of Medicine and Pharmacy of Craiova, 200349 Craiova, Romania; 4Department of Dermatology, Victor Babes University of Medicine and Pharmacy, 300041 Timisoara, Romania

**Keywords:** desmoplastic melanoma, histopathological subtype, diagnosis, prognostic, dermoscopy, reflectance confocal microscopy

## Abstract

Desmoplastic melanoma accounts for 5% of all cases of melanoma, but its diagnosis can be difficult due to its frequent clinical presentation with amelanotic lesions. Histologically, spindled melanocytes surrounded by a collagenous stroma are observed. Compared with other types of melanoma, the desmoplastic types presents greater local aggression, and is more prone to local recurrence, but has a lower risk of lymph node metastasis. Early detection, accurate staging, and proper surgical management are the main factors associated with higher survival rates in melanoma patients. Reflectance confocal microscopy (RCM) has proven to be a valuable imaging tool in the diagnosis of skin neoplasms, being useful for orientating practitioners towards the diagnosis of melanoma and indicating the necessity of performing a diagnostic biopsy. We present the case of 52-year-old woman, who presented to the dermatology department with an irregular, dark-colored plaque in the right deltoid region. Dermoscopy showed asymmetry with an atypical network and some areas of regression. RCM revealed pagetoid cells in the upper epidermis, cell atypia, non-edged papillae, dermal inflammation, and nucleated cells in the dermis, which are highly suggestive of melanoma. A biopsy was also performed. A histopathology exam confirmed the diagnosis of superficially spreading melanoma with a desmoplastic component, and revealed a Breslow index of 0.9 mm, Clark level IV, an absence of mitoses, angiolymphatic invasion and regression, and complete excision. The CT and PET-CT scans were negative. A biopsy of the axillary sentinel lymph node was conducted, with a negative result obtained, establishing the IB stage of the disease. The patient will remain under follow-up to look for a recurrence or a new primary melanoma.

## 1. Introduction

Melanoma is a malignant tumor of the skin that occurs due to abnormalities in the growth and differentiation of melanocytes. Cutaneous melanoma can develop in a previously existing lesion or on apparently normal skin. Multiple risk factors have been proposed to be involved in the onset of melanoma (family history, immunosuppression, number of pre-existing nevi, etc.), but UV radiation exposure, particularly UV-B sunlight in an intermittent high manner, is considered to be the most important one. The incidence of melanoma has risen over the past decades, with the highest prevalence being observed among white males and at a mean age of diagnosis of 65 [1]. Despite accounting for approximately 4% of all skin malignancies, melanoma is responsible for roughly 80% of skin cancer-related deaths [1,2]. Even though there have been advances in the detection of the disease, the overall mortality rate due to melanoma is higher because of the increase in the incidence [3]. Proper management includes early detection, accurate staging, and surgical resection of the tumor, which have been associated with a higher chance of survival in melanoma [1,3,4]. The prognosis is variable and is influenced by numerous factors such as histological parameters (depth of tissue invasion, presence/absence of ulceration, and mitotic rate), age and sex, location (trunk, head, and neck melanomas), and lymph node status at the time of the diagnosis [4,5].

Desmoplastic melanoma (DM) is a relatively uncommon form of melanoma that makes up less than 5% of all cases of melanomas. DM is most commonly found on sun-damaged skin, particularly on the head and neck, with the second most frequent location being the trunk and extremities [6]. Men are affected twice as frequently as women are, and the median age is 60 years. Clinical diagnosis can be difficult because it is more frequently amelanotic, presenting as a nodule, plaque, or scar-like lesion. Histologically, it is characterized by the presence of spindle cells isolated by a fibrous collagen matrix [6,7]. Particularly, nerve involvement can be observed (neurotropism) and is associated with deep-infiltrating tumors [8]. Compared with other types of melanoma, DM exhibits significantly greater local aggression, is infiltrative and prone to local recurrence, and has a lower risk of lymph node metastasis [9]. Memorial Sloan-Kettering Cancer Center (MSKCC) proposed a classification of DM, pure DM (pDM) and mixed DM (mDM), based on the proportion of desmoplasia detected in the sample. For pDM, the desmoplastic component represents >90% of the invasive tumor, whereas in the mDM, cellular areas accounted for 10% to 90%, mixed with collagenous stroma [6,9]. This classification has an impact on the clinical presentation and also the clinical outcome, with mDM being associated with an increased risk of lymph node metastasis and a poorer prognosis compared with that for pDM [8,9].

In dermatology, in vivo reflectance confocal microscopy (RCM) is a valuable imaging technique. It offers a non-invasive, histological-like view of the skin, revealing anatomical structures as well as individual cells. It is particularly useful in the diagnosis of skin neoplasms, mainly because melanin can be visualized as hyperreflective and provides strong contrasted images [10]. Also, it can be a valuable tool in the pre-operative assessment of tumor margins in skin neoplasms, with its highest applicability in basal cell carcinoma [11]. Although it is difficult to assess the histological subtype via RCM examination, this technique remains very useful in orientating the clinician towards a diagnosis of melanoma in the case that patients refuse a biopsy in the first place. Typically, the DM characteristics are similar to those of melanoma, including pagetoid cells and cellular atypia. Other features commonly found are dermal spindle cells and nucleated cells along with dermal inflammation [12,13]. An observation of those features in an RCM examination is a strong indicator of melanoma and of the necessity of performing a diagnostic biopsy.

## 2. Case Presentation

We present the case of a 52-year-old non-smoking woman who presented to the dermatology clinic with a 1.5 cm oval lesion in the right deltoid region with irregular edges and a dark-brown color (Figure 1). She had no prior underlying medical conditions and no family history of skin cancer. The patient first noticed the lesion several years ago with a recent evolution in size, developing a nodule-like, discretely elevated structure in the lower part, and a slightly changed appearance in the last two years. Dermoscopy showed asymmetry of color, shape, and structure with an atypical network and some areas of regression, and a pigmented astructural, peripheral area (Figure 2). The dysplastic nevi and melanoma diagnosis were taken into consideration as differential diagnoses.

The patient preferred a non-invasive technique rather than a biopsy at first, so we decided to perform in vivo RCM examination (with the Vivascope 1500/3000 confocal microscope-Vivascope GmbH, Munich, Germany), which revealed pagetoid cells spreading in the upper epidermis, cell atypia, non-edged papillae, dermal inflammation, and nucleated cells in the dermis (Figure 3A), suggesting the diagnosis of melanoma.

Consecutively, taking into consideration the size and location of the lesion, and the RCM aspect highly suggestive for melanoma, the invasive procedure could not be avoided anymore and an excisional biopsy was performed. Histopathological examination revealed that medium–large melanocytic proliferation developed at the dermo-epidermal junction, with a flat, asymmetrical, imprecisely circumscribed silhouette composed of atypical melanocytes arranged in a lentiginous pattern, sometimes continuous, with the pagetoid invasion of the epidermis and formation of anisomorphic nests. In addition, hyperplasia and basal hyperpigmentation were found in the epidermis, and the underlying dermis had a dense lympho-histiocytic infiltrate (Figure 4). For a better assessment of tumor thickness and a clear distinction from other histologic mimics, an immunohistochemical study was performed, which revealed positive markers: SOX10 and Melan A (Figure 5).

The histological and immunohistochemical examinations confirmed the diagnosis of superficial spreading melanoma with a desmoplastic component, a Breslow index 0.9 mm, Clark level IV, an absence of mitoses, angiolymphatic invasion, and regression.

A clinical examination and an ultrasound of the loco-regional lymph nodes were performed with negative results. The examination for micrometastasis was negative. Further imaging examinations, such as CT for the thorax, abdomen, and pelvis, and PET-CT, were negative.

Considering the thickness of 0.9 mm and the patient’s young age, we decided to perform a sentinel lymph node biopsy (SLNB), with the patient being referred to the plastic surgery department. Preoperative mapping using Tc-99m injected in the perilesional biopsied area was carried out, followed by lymphoscintigraphy, which determined the sentinel node. Following the SLNB procedure, wide local excision with 1 cm margins was performed along with deep excision reaching the fascia. The final diagnosis stage was established as IB a in accordance with the AJCC 8th TNM classification [14].

The patient will remain under follow-up to look for a recurrence or a new primary melanoma. Whole-body photography and digital dermatoscopy for sequential examinations were performed. During the first three years, she will undergo clinical dermatological examinations every three to six months, with a frequency of every six months for the next four to ten years, then annually for life. Regarding the detection of sub-clinical nodal disease, she was recommended to perform lymph node sonography every six months in the next three years. Laboratory examinations such as LDH and S-100 can be performed every six months in the first three years.

## 3. Discussions

DM is a type of spindle cell melanoma that can occur on its own or in conjunction with other types of melanomas, such as melanoma in situ or lentigo maligna. The median thickness of DM, which is 2–4.4 mm, is higher than that of other types of melanoma (non-DM), due to the difficulties in establishing the initial diagnosis [9]. Although data from the literature are conflicting, it appears that DM has a higher local recurrence rate than non-DM does. Some claim that the rate of local recurrence for DM is generally 20–30%, especially for the pure variant, while others claim that the local recurrence rate is 3–7%, similar to that of non-DM [6,15]. It is reported that DM is associated with less regional lymph node metastasis at the time of diagnosis compared with conventional melanomas [7]. More recent reports indicate that the key prognostic factors regarding the survival of patients were similar for patients with DM compared to those with non-DM [9].

The clinical features of DM can lead to frequent misdiagnosis because it appears as non-pigmented papules, nodules, or plaques, or as poor-defined scar-like lesions [8]. The different diagnoses include dermatofibroma, dermal scar, sarcomas, and benign nervous or even sarcomatoid carcinomas [6,7]. A correct diagnosis must be established by the clinician. Among the non-invasive methods, RCM offers high-resolution skin imaging at a cellular level, allowing the real-time visualization of the epidermis and superficial dermis. RCM features are correlated with dermatoscopic and histologic findings and have been described for pigmented lesions such as nevi and melanoma lesions [12,16]. An algorithm developed by Pellacani et al. consists of six criteria for the diagnosis of melanoma: two major criteria accounting for two points each and four minor criteria with one point each. Major criteria include cytologic atypia at the dermo-epidermal junction and non-edged dermal papillae, while minor criteria include roundish pagetoid cells spreading upward, pagetoid cells spreading throughout the lesion, cerebriform clusters, and nucleated cells in the papillary dermis. A total score equal or greater than three strongly suggests the diagnosis of melanoma, with 97.3% sensitivity and 72.3% specificity [17]. The clinical applications of RCM in melanoma include the diagnosis, assessment of lesions on cosmetically sensitive areas, and guidance of the areas for biopsy or for follow-up. Moreover, RCM represents an useful tool not only in detecting melanoma on new arising lesions, but also in increasing the specificity of monitoring dysplastic nevi and detecting malignant transformation in those lesions [18]. In a study that included 64 lesions, the melanoma diagnosis sensitivity was 100% and the specificity was 69%, which resulted in the possibility of avoiding an unnecessary biopsy of a benign nevus. At the moment, histopathologic examination remains the gold standard for melanoma and should be performed if RCM reveals suggestive characteristics [10,12]. Another study by Maher et al. evaluated the utility of RCM and dermoscopy in establishing the diagnosis of DM. Although common RCM features have been found in DM, superficial spreading melanoma, and non-DM invasive melanoma, it is difficult to establish the desmoplastic type through RCM [19]. Nevertheless, RCM seems to be a useful tool in orientating the diagnosis towards melanoma in the case of amelanotic DM lesions [19]. Although the resolution of the RCM method allows a good evaluation of superficial layers of the skin, the evaluation of deep lesions is difficult, due to a decrease in this parameter for lesions located lower than the reticular dermis. For that, the evaluation of some DMs (which, in general, present a higher median thickness than other melanomas do) may be difficult, with newer methods being necessary in order to also evaluate the thickness of the lesions, without losing the resolution in the superficial layers [20]. Line-field confocal optical coherence tomography (LC-OCT) represents a new technique that combines the principles of RCM and optical coherence tomography, resulting in an examination that allows a good visualization of the lesion in both the horizontal and vertical plane. The RCM criteria for melanoma diagnosis (irregular honeycomb pattern; pagetoid spread) are also found in LC-OCT, with the same histopathological correlation, but reconstruction in vertical, horizontal, and 3D images allows a better evaluation of thick lesions, such as those in DM [21,22].

Many attempts have been made over the years to define diagnostic criteria for DM. Recently, it has been divided into two types, pDM and mDM, as mentioned before. In pDM, fibrosis and desmoplastic components account for over 90% of the tumor, while the desmoplastic component accounts for less than 90% in the mixed form. This distinction is critical because the type of DM affects disease management and prognosis [7,9]. For instance, some studies claim that the thickness of pDM is greater than that of mDM [6]. In a paper by Skelton et al., it was noted that increased cellularity can be associated with more aggressive biologic behavior [23]. It is important to mention that some studies report a higher rate of lymph node metastasis in mDM compared with pDM and that it accounts for 6–17% of cases. The histologic subtype is also a prognostic determinant, with mDM patients having a higher risk of metastases and death compared with pDM patients [6]. The mortality rate is higher for patients with mDM, with a 5-year mortality rate of 31% compared with that of 11% in the pure type [6,24].

In our case, the clinical picture depicted a pigmented lesion, more suggestive of a mixed form. Studies show that mDM subtypes are easier to diagnose as melanoma due to the association with lentigo maligna or superficial spreading melanoma, whereas pDM might be inconspicuous, presenting less epidermal components and pigmentation [8]. Histologically, the desmoplastic component represented less than 90% of the tumoral mass; therefore, it fell under the mDM category. Regarding management, the histological subtype might be useful in indicating future investigations. For instance, SLNB is recommended for patients with mDM, but it is debatable whether or not patients with pDM should undergo the procedure due to the lower metastasis rate [6,25].

In accordance with the current guidelines, DM treatment is similar to that of other types of melanoma and involves surgical excision with a safety margin depending on the tumor’s thickness (in situ melanoma: 0.5 cm margins; melanomas with Breslow: ≤1 mm: 1 cm margins; lesions with Breslow 1.0–2 mm: 1 to 2 cm margins) [3,5,25]. An extensive case series study suggested that for DM, the minimal acceptable margin should be 1 cm, with wide local excision being associated with better prognosis and a lower risk of local recurrences [26]. The indication for performing SLNB depends on national regulations, but it is critical to perform, particularly in the mixed form with a Breslow score of at least 0.8 mm because it has an impact on the disease’s prognosis [6]. For example, lesions with thickness values in the range of 0.75 to 1.49 mm have a 25% risk of regional metastasis, whereas for the lesions with a thickness greater than 4 mm, the incidence goes up to 65%. Therefore, the risk of regional metastasis is directly proportional to tumor thickness. Sentinel node status is important because it represents the most important prognostic factor for disease recurrence and is a predictor for survival in malignant melanoma patients [3,27].

Follow-up strategies aim to identify recurrent disease at an early stage, detect a subsequent secondary melanoma, provide education for the patient and first-degree relatives, and offer psychosocial support. There is no universally recommended follow-up scheme, but the general agreement is that the first five years after the primary excision are the most important [27]. Moreover, the RCM examination proves its utility in the cases of patients with multiple atypical nevi that have already been diagnosed with a melanoma in order to avoid or to decrease the number of unnecessary excisions [10,12]. RCM can represent a useful tool in the follow-up of melanoma patients because of its ability to confirm tumor clearance or tumor recurrence, but further studies are needed in order to implement the method.

## 4. Conclusions

Although rare, DM can be quite problematic if not recognized as such, due to its great local recurrence capacity. Always, DMs should be excised with larger margins. RCM can be a useful tool in assessing the examined melanocytic tumor and can orient practitioners towards a diagnosis of DM, even if they cannot yet distinguish between pDM and mDM.

## Figures and Tables

**Figure 1 life-14-00574-f001:**
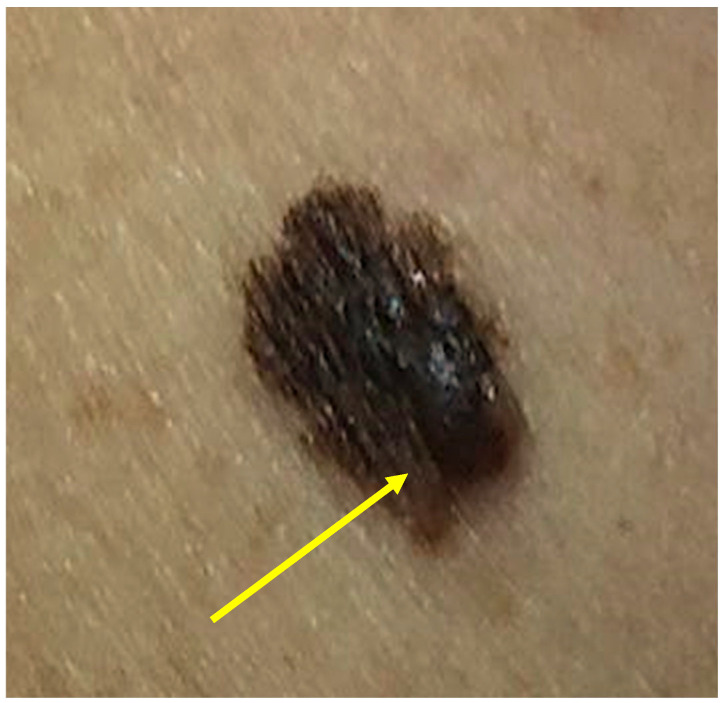
Clinical aspects: slowly growing, brown heterogenous plaque measuring 15 × 12 mm, with imprecise borders and asymmetry, and with a dark-brown papule developed in the inferior part of the lesion (yellow arrow).

**Figure 2 life-14-00574-f002:**
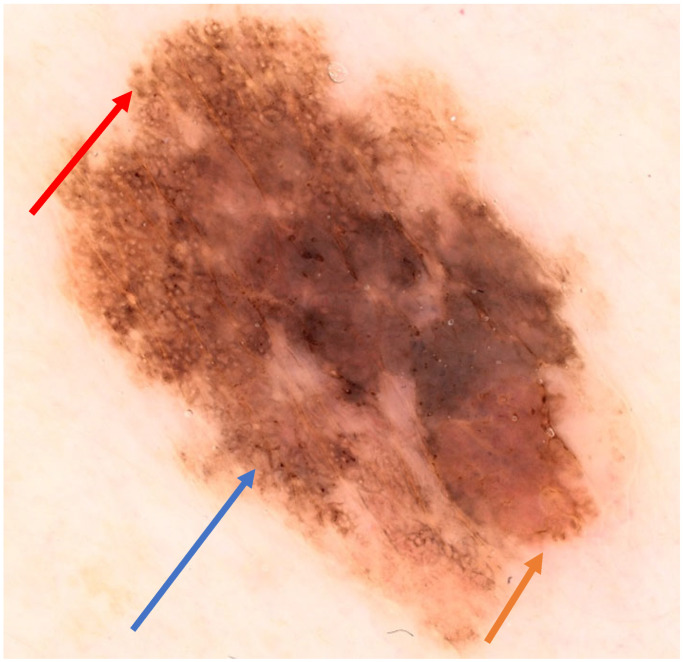
Dermoscopic aspects: asymmetry of pattern and colors; atypical pigment network; abrupt ending of the pigment network in the superior part of the lesion (red arrow); angulated lines (blue arrow); pseudopodes (orange arrow); central area with pigment blotches that alternate with fibrose-like discoloration.

**Figure 3 life-14-00574-f003:**
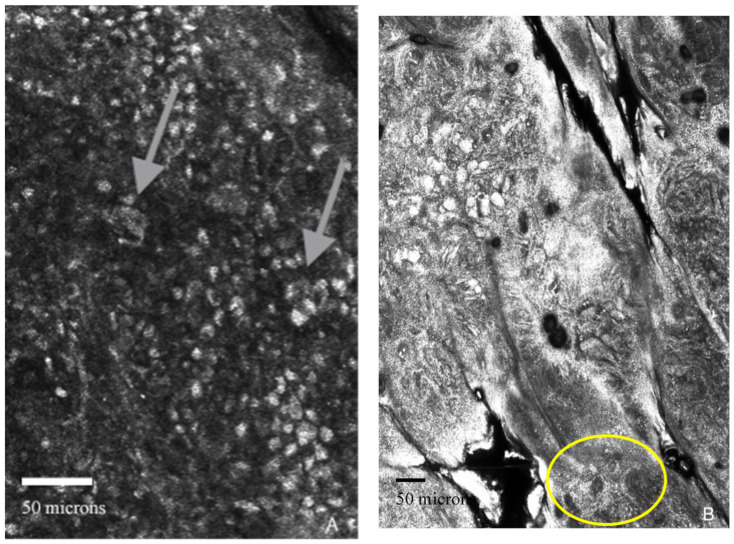
(**A**–**D**). Reflectance confocal microscopy aspects (VivaScope 1500/3000). (**A**) Arrows: pagetoid cells at the level of the epidermis; (**B**) yellow circle: small hyper reflective inflammatory cells; (**C**) yellow square: cerebriform nest; (**D**) yellow circle: disarray of dermal papillae.

**Figure 4 life-14-00574-f004:**
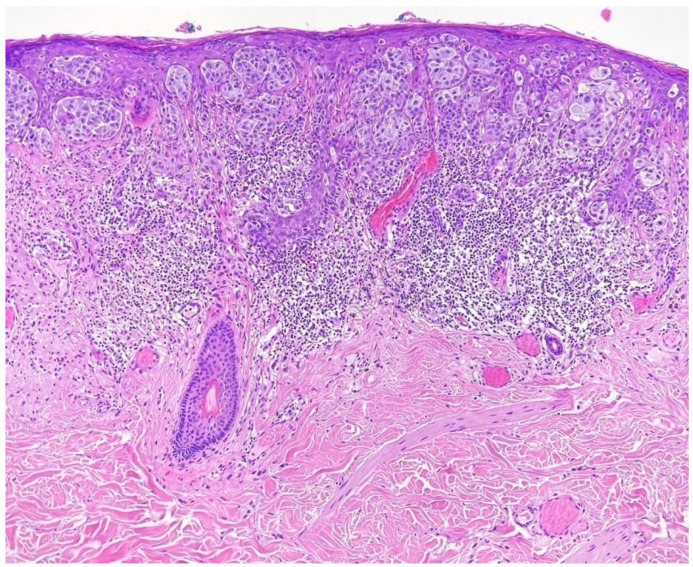
Histological examination, hematoxylin–eosin stain (10×): melanocyte proliferation at the dermo-epidermal junction, atypical melanocytes arranged lentiginously and continuously; pagetoid invasion of the epidermis and formation of anisomorphic nests; lympho-histiocytic dermal infiltrate.

**Figure 5 life-14-00574-f005:**
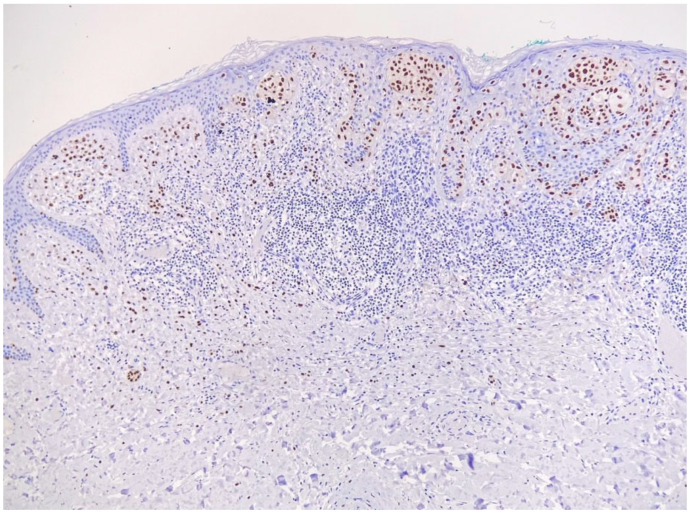
Immunohistochemical expression of SOX 10 in the melanocytic lesion (4×).

## Data Availability

No new data were created or analyzed in this study. Data sharing is not applicable to this article.

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
