# Peer review of "The Importance of In Vivo Reflectance Confocal Microscopy in a Case of Desmoplastic Melanoma"

_life, 2024, doi:10.3390/life14050574_

Round 1
Reviewer 1 Report
Comments and Suggestions for Authors
A nice work on confocal microscopy and desmoplastic melanoma, it could be useful to add some info about LC-OCT. Moreover I suggest to add more info on confocal features of melanoma.
I suggest this work
Licata G, Scharf C, Ronchi A, Pellerone S, Argenziano G, Verolino P, Moscarella E. Diagnosis and Management of Melanoma of the Scalp: A Review of the Literature. Clin Cosmet Investig Dermatol. 2021 Oct 7;14:1435-1447. doi: 10.2147/CCID.S293115. PMID: 34675579; PMCID: PMC8504470.
Author Response
We acknowledge #Reviewer 1 for the comments that definitely helped us to improve the manuscript.
Thank you for this suggestion, we cited and discussed the suggested paper. Also, we have added a paragraph about the LC-OCT.
Reviewer 2 Report
Comments and Suggestions for Authors
A very well written and researched background for the case report.
The significance of the content can be improved substantially:
[1] It seems that there was a management dilemma, in the sense that the patient was reluctant to undergo biopsy, given the remark “patient preferred a non-invasive technique rather than a biopsy in the first place”.
[2] However, given that the case report is about the confocal microscopy features, it would be advantageous to expand on any diagnostic dilemma faced after physical examination and dermatoscopy in this particular patient.
[3] Then, there needs to be a narrative about how such a diagnostic dilemma was resolved by using confocal microscopy, followed by how such information changed the management of the lesion in this particular patient.
Author Response
We acknowledge #Reviewer 2 for the comments that definitely helped us to improve the manuscript.
Thank you for your observations and suggestions. We have made the changes accordingly in order for the text to be clearer.